# How Does Parental Early Maladaptive Schema Affect Adolescents’ Social Adaptation? Based on the Perspective of Intergenerational Transmission

**DOI:** 10.3390/bs14100928

**Published:** 2024-10-10

**Authors:** Ying Shi, I-Jun Chen, Mengping Yang, Liling Wang, Yunping Song, Zhiyin Sun

**Affiliations:** Department of Psychology, Soochow University School of Education, Suzhou Industrial Park (SIP), Suzhou 215121, China; shiying@stu.suda.edu.cn (Y.S.); yangmengping@szys.net (M.Y.); llwang@stu.suda.edu.cn (L.W.); ypsong@stu.suda.edu.cn (Y.S.); zysun@stu.suda.edu.cn (Z.S.)

**Keywords:** early maladaptive schema, social adaptation, intergenerational transmission

## Abstract

An individual’s social adaptation is affected by their early maladaptive schemas. Previous studies have shown that early maladaptive schemas may be intergenerationally transmitted in families. It is important to explore the intergenerational effect of early maladaptive schemas on adolescents’ social adaptation, as they are in a critical period of growth and development. In this study, a cross-sectional design and questionnaire survey were used to collect data to explore the intergenerational influence of early maladaptive schemas in families and their relationship with adolescents’ social adaptation. The participants were 201 adolescents aged 12 to 16 years and their primary caregivers (father or mother), of whom 125 (62.2%) were boys and 76 (37.8%) were girls. There were 70 fathers (34.8%) and 131 mothers (65.2%). Chinese adolescents and their primary caregivers were surveyed using paired questionnaires, and the Young Schema Questionnaire (short form) and Adolescent Social Adaptation Scale were completed. The results show that adolescents’ early maladaptive schema plays an intermediary role between parents’ early maladaptive schema and adolescents’ social adaptation. Parental mistrust/abuse and insufficient self-control schemas affected adolescents’ social adaptation through the mediating effect of their corresponding schemas. Our results reveal the negative impact path of parents’ early maladaptive schemas on adolescents’ social adaptation and provide a new direction for the clinical practice of adolescent family therapy.

## 1. Introduction

Social adaptation refers to a state of a harmonious balance between individuals and the social environment by adjusting their physical and mental state through continuous interaction with the living environment [1]. It is the development task of an individual’s whole life, an important target of individual growth, and a key indicator of physical and mental health development. From a psychological perspective, social adaptation involves the development of individual coping mechanisms and strategies that enable individuals to manage social challenges and stressors. This can include learning social norms, developing interpersonal skills, and adjusting one’s behavior to conform to social expectations [2]. Adolescence is a critical period for the development of individual social adaptation. However, adolescents at this stage are affected by external factors such as increased academic pressure [3,4] and poor parent–child communication [5], and they are easily affected by cognitive structures related to adverse childhood experiences; the possibility of maladaptive behaviors increases geometrically with the rise of grades [6], which has attracted extensive attention from families, schools, and society. In order to implement effective prevention strategies and treatment interventions for socially maladjusted youth groups, it is necessary to identify the impact on the families on which they depend, especially on their parents. Early maladaptive schemas (EMSs), as a kind of rigid cognitive structure and dysfunctional belief system related to an individual’s adverse childhood experience, of the adolescents may be influenced by their parents [7,8,9], and then, this may have an impact on adolescents’ social adaptation.

The aim of this study was to examine the relationship between EMSs and adolescents’ social adaptation by exploring the intergenerational influence of EMSs. We tested whether there is an intergenerational transmission effect of EMSs, and we also tested whether parental EMSs have an impact on adolescents’ social adaptation through their adolescents’ EMSs as a mediator.

### 1.1. Early Maladaptive Schema and Social Adaptation

The concept of EMSs proposed by Young [10] divides the negative mental models formed by individuals who do not receive core emotional needs from caregivers in early life into five schema categories: (1) separation and rejection, including abandonment/instability, mistrust/abuse, emotional deprivation, defectiveness/shame, and social isolation; (2) insufficient autonomy and performance, including dependence/incompetence, vulnerability to harm or illness, and enmeshment/undeveloped self and failure; (3) impaired limits, including grandiosity and insufficient self-control; (4) other-directedness, including subjugation and self-sacrifice; and (5) over vigilance and inhibition, including emotional inhibition and unrelenting standards/hypercriticalness [11]. Once formed, EMSs are stored in the form of emotional memory, physical feeling, and behavioral response and become a stable mode. When experiencing similar traumatic events, EMSs will be activated, and the individual’s body and mind will be affected by self-defeating cognitive styles, negative emotions and somatic feelings, directly or indirectly affecting social adaptation, which has a significant impact on an individual’s lifetime [12].

EMSs can affect many aspects of adolescents’ social adaptation. For example, the more obvious the EMS is, the more obvious the procrastination behavior is [13] and the lower the academic achievement is [14]. This may be because individuals have formed EMSs (e.g., “I am incompetent” and “I am worthless”) due to failures experienced in their early years that affect their sense of self-efficacy and thus their academic performance [15]. In addition, almost all EMSs (e.g., mistrust/abuse, social isolation/alienation, harsh standards/demanding) are significantly related to adolescents’ interpersonal communication problems [16,17] because EMSs can enable individuals to assume ways of interacting with others based on past adverse experiences; for example, individuals whose schemas tell them “others cannot be trusted” and “I am not worthy of love” plan their interpersonal behavior in advance [18] and cannot establish safe and satisfactory relationships with others.

Parental EMSs may similarly have an impact on adolescent social adaptation, as dysfunctional parental beliefs determine their approach to functioning in family relationships (i.e., showing negative parenting characteristics like hostility, abuse, indifference, rejection, etc.), affecting the quality of parent–child interaction. In China, due to the conflict between traditional culture and modern thought, the role of parents in the family is changing. To prevent their adolescents from “losing at the starting line”, parents have invested more money and energy than ever before in their adolescents’ education (especially academic achievement). They have high expectations and strict requirements for their adolescents but know little about their adolescents’ spiritual needs [19], making them unable to meet their adolescents’ core emotional needs [20] and affecting their adolescents’ social adaptation. Therefore, for adolescents who are in a critical period of individual growth and development, exploring EMSs’ impacts on their social adaptation can provide an early warning and help them avoid developing more serious pathological problems.

### 1.2. The Influence of Intergenerational Relationship of EMSs on Adolescents’ Social Adaptation

Empirical studies have found that early maladaptive schemas are highly stable and lasting beliefs [21,22], just as human values and social attitudes may be transmitted generationally [7,8,9]. This is due to the fact that with the birth of a child, the EMSs of parents may be activated by parent–child conflict in the process of child care, leading to stereotypical and dysfunctional parenting behavior patterns, increasing parental pressure, reducing parenting sensitivity, and therefore failing to meet the emotional needs of their adolescents [23].

Given that EMSs are negative and stable assumptions about oneself and their relationships with others formed during childhood and adolescence [24], adolescence may be a critical period for the formation of EMSs. During this period, adolescents’ increased independence and autonomy may bring about changes in their emotional needs for their parents [25]. If parents still adopt negative emotions and negative interactive behaviors to treat adolescents under the influence of their own EMSs, adolescents will show more emotional disorders and problem behaviors [26], resulting in an increased risk of maladjustment. However, to our knowledge, no study has examined the association between parental EMSs, adolescents’ EMSs, and adolescents’ social adaptation. It is worth exploring further whether parental EMSs can cause deviations in the cognitive structure, emotion, and behavior of adolescents, thus affecting adolescents’ social adaptation.

In traditional Chinese culture, the family is regarded as the core unit of society, while filial piety and family harmony in Confucianism have a profound impact on the interaction patterns among family members [27]. This cultural perception is particularly evident in parent–adolescent relationships [28], such as respecting parents and elders and fulfilling obligations to care for the family. Studies have also found that Chinese adolescents show higher obedience to their parents [29,30,31]. Therefore, Chinese adolescents may be more likely to accept and be influenced by their parents’ upbringing; that is, the intergenerational transmission of early adaptive schemas occurs. In addition, Chinese parents are less likely to directly show and express love [32] and less likely to emphasize autonomy [33]. This has also led to a higher acceptance of excessive control behaviors, such as corporal punishment [34] and strict supervision [35], among Chinese adolescents. At the same time, parental psychological control, as a specific manifestation of EMSs in the process of parenting adolescents, can adversely affect adolescents’ autonomy and academic achievement and significantly increase their risk of psychopathological symptoms [36,37]. Therefore, this study aims to explore how the EMSs of parents and adolescents interact in the Chinese context from a cultural perspective, and how these factors affect adolescents’ social adaptation. Through this study, we hope to reveal how family care practices unique to China shape adolescent adaptation patterns and provide cultural background support for the formulation of relevant intervention strategies.

### 1.3. Present Study

The purpose of this study is to examine the relationship between parents’ EMSs, adolescents’ EMSs, and adolescents’ social adaptation and to fill the gap of empirical research on how EMSs are passed from generation to generation in families and how these EMSs specifically affect adolescents’ social adaptation. It is expected to provide a new direction for the clinical practice of adolescent family therapy, which can help professionals better understand and intervene in the social adaptation problems of adolescents. The study predicted that (1) early maladaptive schemas may be transmitted in the family; (2) parents’ EMSs influence adolescents’ social adaptation through a mediating role.

## 2. Methods

### 2.1. Participants and Procedures

The participants were middle school students and their primary caregivers (father or mother). The adolescents (*n* = 201) were between 12 and 16 years old (M = 13.91, SD = 1.57), including 125 boys (62.2%) and 76 girls (37.8%). Seventy (34.8%) were fathers and one hundred thirty-one (65.2%) were mothers. The adolescent–parent pairs included both adolescent–father and adolescent–mother combinations. Forty-eight percent of participants were from Grade 7, twenty-six percent from Grade 8, thirteen percent from Grade 9, six percent from Grade 10, five percent from Grade 11, and two percent from Grade 12. The majority of young people come from rural families (63%). Approximately 21% of participants reported a family monthly income of less than ¥4800 ($687), while 45% had incomes between ¥4800 ($687) and ¥9600 ($1373), 17% between ¥9601 ($1374) and ¥14,400 ($2061), and 17% above ¥14,400 ($2061). The parents’ education level was below elementary school (6.0%), middle school (43.9%), and high school or technical school (31.1%). Those with a college degree or above accounted for 19.0%. Of all the families, 165 (82%) were two-parent families, and 36 (18%) were single-parent families.

The study was conducted in Suzhou City, China. We contacted the ethics director of Suzhou Middle School with the help of the local education bureau and sent them an email application to organize student participation in the investigation. A total of 15 schools accepted the invitation. After the ethics officer’s comments, we held a parents’ meeting in the school. During the meeting, parents were introduced to the purpose and procedure of the study, the principle of confidentiality, and the participants’ right to terminate the study. During the meeting, the Internet link of the questionnaire was published, and students and parents who agreed to participate in the questionnaire were willing to click the link and fill in the questionnaire. In order to achieve a matching effect between parents and children, parents and children can use their real names or nicknames when answering questions, but they must ensure that both use the same name. The family’s primary caregiver (father or mother) completed the Young Schema Questionnaire for the parents. Adolescents filled in the Young Schema Questionnaire and the Adolescent Social Adaptation Scale simultaneously. After the parents and adolescents had submitted their answers, the researchers received the results through Questionnaire Star (a professional online questionnaire, evaluation, and voting platform) and then screened and matched the data. The questionnaire was published during the parents’ meeting. In order to avoid the group pressure of parents to participate in the survey, the researcher published two two-dimensional codes at the same time; two-dimensional code 1 was the parents’ questionnaire for the research, two-dimensional code 2 was the teenagers’ questionnaire for the research, and two-dimensional code 3 was the reading materials of public news. Parents who were willing to participate scanned the QR code1: pay attention to the answers, and parents who were unwilling to participate scanned the QR code3: read some public news. After, the parents filled in the questionnaire, handed the mobile phone to their children, scanned and completed the questionnaire or read the news materials. All questions contained in the electronic questionnaire were required; that is, all questions had to be completed before submitting the questionnaire so as to ensure the integrity of the questionnaire. The questionnaire lasted 15 minutes. Respondents did not receive any compensation for participating in the study. The study was approved by the ethics committee of Suzhou University (Approval number: ECSU-2019000189).

### 2.2. Measures

*Young Schema Questionnaire (short form)* [38] was used to measure the EMSs of adolescents and their primary caregivers. It consisted of 75 items that assess 15 schemas covering five schema domains on a 6-point scale from 1 (extreme non-conformity) to 6 (extreme conformity). There were five items per schema scale, giving a possible score range of 5–30 for each factor, with higher scores indicating more obvious EMSs. The scale had a Cronbach α coefficient of 0.97 and a retest reliability of 0.90, proving the questionnaire had sufficient reliability and validity.

*Adolescent Social Adaptation Scale* [39] was used to measure the social adaptation of adolescents. The scale included four dimensions: interpersonal relationship (e.g., I am good at cooperating with my classmates), academic achievement (e.g., I have my own learning plan and goals), life skills (e.g., I can clean my room by myself), and psychological resources (e.g., I can forget sad things quickly). Adolescents answer 33 questions using a five-point scoring scale ranging from “never” to “always”. The average score of all items in the scale represents the degree of social adaptation, with higher scores indicating better social adaptation. The dimensions’ Cronbach’s α coefficient ranged from 0.88 to 0.91.

### 2.3. Data Analytic Strategy

All analyses were conducted in IBM SPSS21.0. The common method bias test was carried out using the Harman single-factor method. After factor analysis, 34 factors with eigenvalues greater than 1 were obtained. The first factor explained 25.47% of the variation, far less than the critical value of 40%, indicating no serious common method bias in this study. First, based on the study purpose, Pearson’s correlation was used to determine whether parents’ and adolescents’ EMSs were related to the adolescents’ social adaptation results. Second, paired t-test was used to investigate the differences in EMSs between parents and adolescents. Third, we referred to the mediation effect test method proposed by Preacher and Hayes’ Bootstrap program [40]. Using the Bootstrap method in the SPSS Process plug-in, the mediation analysis was conducted with the total score of parents’ EMSs as the independent variable, the total score of adolescents’ EMSs as the mediating variable, and the total score of adolescents’ social adjustment as the dependent variable. Fourth, by further using regression, the dimensions of parents’ EMSs were divided into independent variables, the dimensions of adolescents’ EMSs were divided into mediating variables, and the total score of adolescents’ social adaptation was taken as dependent variable to explore the mediating relationship between adolescents’ various types of EMSs and adolescents’ social adaptation.

## 3. Results

### 3.1. The Correlation between Parents’ EMSs, Adolescents’ EMSs, and Adolescents’ Social Adaptation

Descriptive statistics and correlation analysis of parents’ EMSs, adolescents’ EMSs, and adolescents’ social adaptation were carried out to understand their relationship (see Table 1). According to the EMS scores (*M* ± *SD*) of parents and adolescents, compared with parents’ EMS scores, adolescents’ EMS scores of emotional deprivation, social isolation, failure, self-sacrifice, and emotional inhibition were lower, and other EMS scores were higher.

Only items with significant correlation (*p* < 0.05) are displayed in the relevant part to improve readability. The study found a positive correlation between almost all parents’ EMSs and their adolescents’; the higher the parents’ EMSs are, the higher their adolescents’ are and vice versa. Because this study focuses on the intergenerational relationship between parents and adolescents on the same EMSs, the correlation between parents’ EMSs and adolescents’ corresponding EMSs is presented on the diagonal. The results showed that among the 15 EMSs, 7 corresponding EMSs were significantly correlated among parents and adolescents: mistrust/abuse, failure, enmeshment/undeveloped self, self-sacrifice, unrelenting standards/hypercriticalness, grandiosity, and insufficient self-control. This means that parents’ EMSs may directly impact adolescents’ EMSs.

In addition, most adolescents’ EMSs were significantly negatively correlated with their total social adaptation score in all dimensions. In particular, nine EMSs (emotional deprivation, mistrust/abuse, social isolation, defectiveness/shame, failure, dependence/incompetence, enmeshment/undeveloped self, subjugation, and insufficient self-control) were significantly negatively correlated with all dimensions of their social adaptation; the higher their EMS score is, the lower their level of social adaptation is. However, there was a significant positive correlation between self-sacrifice EMSs and all dimensions of social adaptation; the higher their self-sacrifice schema score is, the higher their level of social adaptation is. In addition, there was no significant correlation between unrelenting standards/hypercriticalness or grandiosity EMSs and social adaptation. As this study found no significant correlation between parents’ EMSs and their adolescents’ social adaptation, it is not presented in the table.

### 3.2. Parent–Adolescent EMSs Intergenerational Relationship Analysis

Paired sample *t*-tests were used to test the consistency between parents’ and adolescents’ EMSs. The results (see Table 2) showed significant differences between parents and adolescents in such EMSs dimensions as emotional deprivation, abandonment/instability, social isolation, defectiveness/shame, dependence/incompetence, vulnerability to harm or illness, enmeshment/undeveloped self, unrelenting standards/hypercriticalness, and grandiosity (*p* < 0.05). Specifically, adolescents’ scores were significantly higher than parents’ for all EMS except emotional deprivation, where they were significantly lower. There was no significant difference in parents’ and adolescents’ scores in such EMS dimensions as mistrust/abuse, failure, subjugation, self-sacrifice, emotional inhibition, and insufficient self-control (*p* > 0.05), indicating that there may be an intergenerational transmission effect.

This study conducted an independent sample *t*-test on father–adolescent and mother–adolescent EMSs to explore the differences in parent–adolescent EMS relationships when the primary caregiver was the father or mother, respectively. The results showed that adolescents’ EMS scores for vulnerability to harm or illness and enmeshment/undeveloped self were significantly higher (*p* < 0.05) when their father was the primary caregiver; there were no significant differences in other EMSs. When the mother was the primary caregiver, there were significant differences in seven EMSs between the mother and the child (*p* < 0.05), including emotional deprivation, abandonment/instability, mistrust/abuse, social isolation, vulnerability to harm or illness, enmeshment/undeveloped self, and unrelenting standards/hypercriticalness; there were no significant differences in other EMSs. Adolescents’ EMS scores were significantly higher than mothers in all dimensions except emotional deprivation, where they were significantly lower. There was more intergenerational consistency in parent–child EMSs when fathers raised their adolescents, whereas mothers’ EMSs showed more differences between generations.

### 3.3. The Mediating Role of Adolescents’ EMSs in the Influence of Parents’ EMSs on Adolescents’ Social Adaptation

We investigated the possible predictive relationship and mediating effect of adolescents’ EMSs on parental EMSs and adolescents’ social adaptation using the Bootstrap method proposed by Preacher and Hayes, with parents’ EMSs as the independent variable, adolescents’ EMSs as the mediating variable, and adolescents’ social adaptation as a dependent variable [41].

#### 3.3.1. The Mediating Role of Adolescents’ EMSs in Parents’ EMSs and Adolescents’ Social Adaptation

In this study, the total scores of adolescents’ and parents’ EMSs were calculated, respectively, and then their mediating effect on adolescents’ social adaptation was analyzed. The results showed that parents’ EMSs could significantly positively predict adolescents’ EMSs, indicating an intergenerational transmission effect of EMSs. When parents’ and adolescents’ EMSs were entered into the regression equation, the direct predictive effect of parents’ EMSs on adolescents’ social adaptation was not significant, while adolescents’ EMSs could significantly positively predict their social adaptation, and the Bootstrap confidence interval of the mediation effect did not include 0 ([−0.12, −0.01]). Therefore, adolescents’ EMSs had a significant, fully mediating effect on the relationship between parents’ EMSs and adolescents’ social adaptation (as shown in Table 3). The specific path coefficients between variables in this study are shown in Figure 1 based on the analysis of the mediation effect.

#### 3.3.2. The Mediating Role of Each Dimension of Adolescents’ EMSs in the Influence of Each Dimension of Parents’ EMSs on Adolescents’ Social Adaptation

Although the relationship between parents’ and adolescents’ EMSs and adolescents’ social adaptation is clear, the influence of various EMSs dimensions remains unclear. Therefore, this study used a stepwise regression to analyze the mediating effect of each dimension of parents’ EMSs on adolescents’ social adaptation. The results showed that the mistrust/abuse in parental disconnection and rejection EMS domains could significantly negatively predict adolescents’ social adaptation through adolescents’ mistrust/abuse EMSs (*β* = 0.23, *t* = 2.56, *p* < 0.05). In the impaired limits schema domain, parents’ insufficient self-control/self-discipline EMS could significantly negatively predict adolescents’ social adaptation, mediated by the insufficient self-control/self-discipline EMS (*β* = 0.30, *t* = 3.05, *p* < 0.01); other EMSs had no mediating effect. When parental mistrust/abuse and insufficient self-control/self-discipline EMSs were entered into the regression equation, they had no significant direct predictive effect on adolescents’ social adaptation; however, adolescents’ mistrust/abuse and insufficient self-control/self-discipline EMSs could significantly negatively predict adolescents’ social adaptation. Therefore, adolescents’ mistrust/abuse and insufficient self-control/self-discipline EMSs had a significant, fully mediating effect on the relationship between parental mistrust/abuse and insufficient self-control/self-discipline EMSs and adolescents’ social adaptation (as shown in Table 4 and Table 5).

## 4. Discussion

In this study, parents and adolescents were paired in a cross-sectional study to evaluate whether there was an intergenerational transmission relationship between parents’ and adolescents’ EMSs and whether adolescents’ EMSs mediated the relationship between parents’ EMSs and adolescents’ social adaptation. We found an intergenerational transmission effect for EMSs in Chinese families, a significant correlation between parents’ and adolescents’ EMSs, and that partial parents’ EMSs could predict the corresponding adolescents’ EMSs. Parents’ EMSs did not directly affect adolescents’ social adaptation, but they did affect it through the mediating role of adolescents’ EMSs.

### 4.1. The Relationship between Parents’ and Adolescents’ EMSs

The adolescent stage is relatively unstable and easy to change in the individual life with obvious transition. At this stage, the individual’s physiology is basically mature, but the psychological and social behavior is in the early stage of development. They are faced with the task of establishing identity and developing independence and autonomy, and the desire to be independent and free from adult control is particularly strong, but they still need to maintain the attachment relationship with their parents, and the peer relationship is gradually developed [42]. Independence and attachment have become important contradictions in their development, and the resulting conflict often leads to social adaptation difficulties for adolescents. Previous research results have emphasized the importance of parental attachment style, especially anxiety related to attachment. Importance in predicting the quality of relationships formed between parents and their early adolescent adolescents [43].

Most parents’ and adolescents’ EMSs were correlated. Focusing on and comparing parents and adolescents with the same type of EMSs revealed significant correlations between parents and adolescents in seven schemas: mistrust/abuse, failure, enmeshment/undeveloped self, self-sacrifice, unrelenting standards/hypercriticalness, grandiosity, and insufficient self-control; eight other schemas did not show significant correlation. A paired *t*-test conducted on parents’ and adolescents’ EMSs revealed that four of the seven schemas with a significant correlation (mistrust/abuse, failure, self-sacrifice, insufficient self-control/self-discipline) showed no significant difference, indicating consistency between generations.

Specifically, individuals whose parents abused them in childhood may develop a cognitive bias towards aggressive responses; that is, they may believe that an aggressive response is morally acceptable and can produce positive results [44]. Therefore, after becoming parents, they will abuse their adolescents, who will, in turn, develop similar EMSs of distrust of abuse. This is consistent with previous research on the intergenerational transmission of trauma, where grandparents’ experiences of physical and psychological abuse have been found to increase the risk of sexual and physical harm in the parents’ generation, with psychological abuse becoming a particularly critical factor. Not only does it directly impact the parents, but it also increases the risk of sexual victimization for their offspring, thereby exemplifying the familial transmission of psychological trauma. Moreover, this transmission exerts an indirect influence on younger generations, exacerbating their vulnerability to harm and highlighting the intricate complexity of trauma perpetuation across generations [45]. Similarly, if schemas drive parents to believe they were born lacking the qualities needed for success, they will use avoidance as their coping method to avoid developing abilities and taking responsibility and let their adolescents experience failure repeatedly in the child-raising process, forming a failure schema [11].

The intergenerational consistency of the self-sacrifice schema can be explained from the cultural level. In Chinese culture, parental sacrifice has always been regarded as the central feature of the family concept. Adolescents socialized in this culture must make corresponding sacrifices to conform to their parents’ expectations [46]. The influence of parental insufficient self-control/self-discipline EMS on adolescents’ insufficient self-control/self-discipline EMS may be influenced by biological and genetic factors [47] and may be indirectly generated by parental parenting style [48].

This study also found that adolescents’ EMS scores were higher than parents’, indicating adolescents reproduced the EMSs transmitted through parents but to a deeper extent. The results show that if EMSs are not evaluated and repaired in adolescents, their schema activation situation increases as they age, which may cause social maladjustment and even psychological disorders. This study conducted a regression analysis on each schema to explore the influence of parents’ EMSs on their adolescents’ and found that parents’ EMSs, such as mistrust/abuse and insufficient self-control/self-discipline, could significantly predict adolescents’ corresponding EMSs, indicating the intergenerational transmission of EMSs.

The results of this study are similar to those of Sundag et al. [7] and Gibson et al. [49], who found a correlation between parents’ and adolescents’ EMSs. The difference is that Sundag et al. [7] tested the mediating effect based on the overall average score of parents’ and adolescents’ EMSs and found that the intergenerational transmission of EMSs was caused by parents’ overcompensation coping style and adolescents’ memories of negative parenting styles. This study explored the intergenerational relationship between parents and adolescents for 15 EMSs and found that parents’ mistrust/abuse and insufficient self-control EMSs could significantly predict adolescents’ EMSs. Gibson et al. [49] found that mothers and adult daughters had significant correlations only on three EMSs—social isolation, enmeshment/undeveloped self, and subjugation—while we found significant correlations between parents and adolescents on seven EMSs—mistrust/abuse, insufficient self-control, and five related items. This may be because the adolescents in China were still dependent on their parents and thus more likely to be affected by their parent’s cognitive style, emotional experiences, and interpersonal interactions in their native family [50] and internalize their parents’ EMSs.

We also found that fathers’ and mothers’ influence on their adolescents’ EMSs differed when the main caregiver’s gender was considered. When the father was the main caregiver, the adolescents scored significantly higher than their father in vulnerability to harm or illness and enmeshment/undeveloped self EMSs, while the other thirteen schemas showed no significant difference, showing consistency between generations. When mothers were the main caregiver, there was no significant difference between their and their adolescents’ scores in eight schemas, including self-sacrifice, grandiosity, and insufficient self-control, showing consistency between generations. In other words, the intergenerational transmission of EMSs via fathers and mothers was different, similar to the findings of Mącik et al. [9].

In addition, fathers’ EMSs were more likely to affect their adolescents through intergenerational transmission than mothers’. This seems to differ from the traditional “strict father and loving mother” cognition, in which the mother is the main caregiver and can express her feelings and emotions better, so adolescents feel more secure. Adolescents are expected to have a better relationship with their mother than their father and to be more likely to be affected by mother’s EMSs. Fathers, in their traditional male role, teach adolescents more about discipline and success in all areas of social life [51], treat adolescents more rationally, and play a more important role in forming adolescents’ values [52]. Fathers’ cognition, emotions, and behaviors driven by EMSs are more likely to be identified by adolescents, so their EMSs are more consistent with their adolescents’ EMSs than are their mothers’ EMSs. Also, as primary caregivers, mothers may overcompensate when trying to balance their greater involvement in their adolescents’ social lives with their closer emotional ties [53], alternating between being arbitrary and spoiling their adolescents [54]. However, that kind of love is not necessarily what adolescents really want. Instead, they may feel that their mother’s love is unstable or that she distrusts them, leading them to feel more isolated and vulnerable, want to meet her high expectations, and be too hard on themselves.

In short, in the intergenerational transmission of EMSs, adolescents often develop their EMSs based on their parents’ EMSs, albeit influenced by various factors and the adolescents’ own selections and processing. There are both inheritance and differences between parents and adolescents [55].

### 4.2. The Mediating Effect of Adolescents’ EMSs on Parental EMSs and Adolescents’ Social Adaptation

This study evaluated the relationship between parental EMSs and adolescents’ social adaptation. A correlation analysis showed that parents’ EMSs were not significantly correlated with adolescents’ social adaptation, while adolescents’ EMSs were significantly negatively correlated with most dimensions of social adaptation, similar to other research results [11,56]; only self-sacrifice EMSs were significantly positively correlated with social adaptation. This may be because Chinese traditional culture is collectivist and emphasizes interpersonal self-effacement, cooperation, and self-sacrifice, with group interests preceding individual ones [53]. This prosocial performance enables individuals to adapt to the social environment in terms of academic achievement and interpersonal communication and develop good social skills.

The regression analysis showed that parents’ EMSs could not directly affect adolescents’ social adaptation; however, they could completely mediate it through adolescents’ EMSs. Further investigation found that parental mistrust/abuse and insufficient self-control/self-discipline schemas could completely mediate adolescents’ social adaptation through the child’s same schemas. These results indicate that the self-defeating emotions and cognitive styles formed by parents’ unmet emotional needs do not necessarily directly affect adolescents’ social adaptation, and the influence of parents’ emotional management on adolescents’ social adaptation is realized through the bridge of adolescents’ emotional management.

First, parents’ distrust/abuse schema affects adolescents’ social adaptation through intergenerational transmission. Individuals come into the world eager for positive responses from parents and significant others, thus forming a sense of trust and security [57]. If in developing their relationship, parents refuse their adolescents, fail to respond to their adolescents’ emotional needs for care and comfort, or inflict emotional and physical abuse on them, adolescents will form distrust/abuse schema, making them distrust society [11]. When the stressor activates the schema, individuals with the distrust/abuse schema expect others to hurt, abuse, humiliate, cheat, lie, manipulate or take advantage of them; often experience intentional harm, unfair results, and extreme neglect; and may also feel that they are always deceived or disadvantaged by others. This negative self-cognition will interfere with adolescents’ information processing during periods of stress [58], eventually becoming a vulnerability factor in their social adaptation. Parents’ providing a safe, stable, honest, warm, and caring environment for their adolescents is the cornerstone of adolescents’ healthy development.

Second, the insufficient self-control/self-discipline schema affects adolescents’ social adaptation through intergenerational transmission, consistent with Young et al.’s research results [11]. Doting or indulgent parental education is the main cause of insufficient self-control schema. Individuals who develop such a schema tend to perceive themselves as more privileged than others and exhibit low levels of frustration and impulsive behavior [59]. Other studies have shown that people with limited capacity have problems managing impulses, delaying gratification, establishing internal boundaries, and achieving realistic personal goals due to an inability to withstand stress, self-regulate emotions, and think before acting. They are more likely to cope by avoiding problems [60], leaving them fewer opportunities to reassess and solve problems and affecting their social adaptation. This suggests that lessening EMSs’ impact on social adaptation necessitates enhancing individual self-cognition and control, regulating and dealing with stress and difficult emotions, developing positive coping styles, establishing goals, formulating action plans, controlling behaviors, assuming responsibilities, and developing skills to adapt to society. Further attention should be paid to the negative impact of parents’ lack of self-control/self-discipline schema.

Although this study examined the mediating role of adolescents’ EMSs in parents’ EMSs and adolescents’ social adaptation, only two schemas, mistrust/abuse and insufficient self-control/self-discipline, were analyzed as mediators. That is to say, of the fifteen schemas proposed by Young et al. [11], thirteen did not affect adolescents’ social adaptation through intergenerational transmission, which may be related to the schema theory that individuals may develop overcompensation strategies when schemas are activated [11]. When parents fail to meet a child’s emotional needs, the child compensates in an opposite way to reduce the pain caused by the schema. For example, a highly controlled individual may reject all forms of control as an adult and be indulgent in raising their adolescents, giving them all the freedom and love they want. Therefore, it seems that parents’ EMSs are not passed on to adolescents, but the negative impact of overcompensatory coping styles still exist, which can be further explored in future studies.

Our results provide a significant practical reference for cultivating adolescents’ social adaptation from a family system perspective. We propose intervention ideas for improving the social adaptation of adolescents, specifically as follows:Set up classes for parents to help parents understand the formation of early maladaptive schemas, learn scientific parenting styles, provide more support and care for adolescents, and provide a strong guarantee for adolescents to enhance their social adaptability.Use schema therapy to intervene and improve EMSs in adolescents. For example, the use of image reshaping technology to help adolescents reshape their emotional responses to early negative experiences can effectively change the impact of early bad schemas on adolescents’ emotions and behaviors. Through chair work techniques, adolescents are allowed to engage in dialogue with different parts of themselves, thereby dealing with and changing their early bad schemas, helping adolescents better understand and integrate inner conflicts. These specific strategies not only help adolescents to deal with and improve early maladaptive schemas more effectively and enhance their adaptive ability but also provide practical guidance for clinical work and further optimize intervention effects [61].

Despite the results cited, it is important to highlight limitations in this study that should be considered. First of all, cross-sectional data explore only correlation rather than causation. Therefore, it is suggested that future studies use longitudinal study designs to examine these links. Second, the sample size of this study is small, which may limit the possibility of identifying the universality of and differences in EMS development due to factors such as adolescents’ different grades and family structure backgrounds. Future studies could expand the sample size and use a wider sample of adolescents of different ages to examine EMSs and adolescents’ social adaptation and detect regularity for earlier prevention and intervention. In addition, as family structures may affect EMSs and social adaptation, future studies could expand the sample size to compare and verify the intergenerational transmission of EMSs in single-parent and two-parent families. In addition, although the results show that parents’ EMSs can influence adolescents’ EMSs and thus their social adaptation, family ecology is very complex, and other factors may play a role in the intergenerational transmission of EMSs. For example, co-parenting and intergenerational parenting are increasingly common in Chinese families, and interaction patterns among family members and satisfaction with family relationships may impact adolescents’ cognitive development and social adaptation; all these factors are worthy of further research. Based on this, the host–object interdependence model considers the non-independence of pair data, an emerging method of pair-data analysis in marriage, family, and other research fields, and provides an effective means for interpreting pair relationships. Our study only considered one parent’s influence on adolescents’ EMSs and social adaptation; future research could consider the joint roles played by other important family caregivers and whether adolescents’ upbringing behaviors are not driven only by one parent’s EMSs (as subject effect), but may be affected by other caregivers’ EMSs (as an object effect). Future research could investigate how family members’ interactions influence adolescents more comprehensively.

This study explored how EMSs influence social adaptation from an intergenerational transmission perspective. Adolescents’ EMSs negatively predicted adolescents’ social adaptation. Parental EMSs did not directly negatively affect adolescents’ social adaptation but indirectly negatively affected adolescents’ social adaptation through the intergenerational transmission of EMSs, with adolescents’ EMSs playing a mediating role between parental EMSs and adolescents’ social adaptation. We found that parents’ and adolescents’ EMSs exist through some degree of intergenerational transfer; however, such a transfer is not absolute, even if individual EMSs are relatively stable. However, it is still possible to detect, repair, and block or partially block schemas’ intergenerational influence on adolescents by interacting with family members in the family environment. This finding supports the value and feasibility of schema therapy proposed by Young [11]. EMSs have a persistent negative impact on individuals’ social adaptation and may be passed down through generations. This article discovered that parents’ schemas of mistrust/abuse and insufficient self-control are potential predictors of adolescents forming and keeping EMSs. Moreover, the important role of social adaptation provides a new direction for adolescent family therapy in clinical practice; clinicians could help visitors better understand the deep reasons for social maladjustment, namely bad core beliefs, and their long-term effects. These EMSs are deconstructed and repaired in therapy to improve social adaptability.

## Figures and Tables

**Figure 1 behavsci-14-00928-f001:**
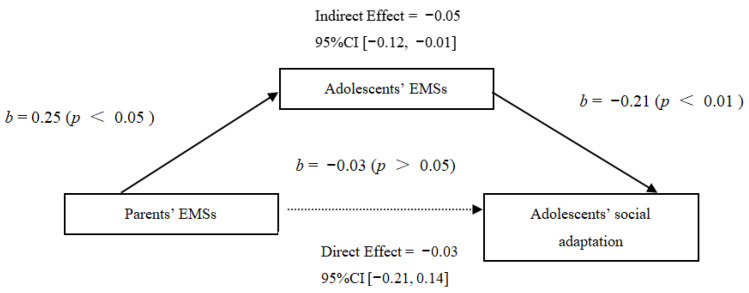
The mediating graph of adolescents’ EMSs between parents’ EMSs and adolescents’ social adaptation. Note: All the variables in the model after standardized processing to the regression equation; the solid line indicates significant coefficient, and the dotted line indicates insignificant coefficient. The same below.

**Figure 2 behavsci-14-00928-f002:**
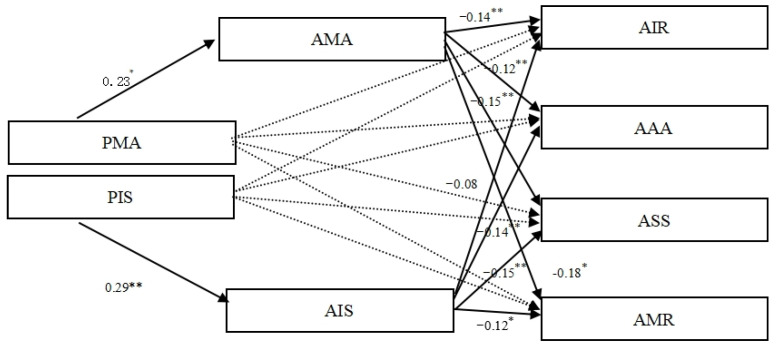
The mediating graph of adolescents’ EMSs between parents’ EMSs and adolescents’ social adaptation. Note: * *p* < 0.05, ** *p* < 0.01.

**Table 1 behavsci-14-00928-t001:** Correlation between parents’ EMSs, adolescents’ EMSs, and adolescents’ social adaptation. (*N* = 201).

Parents’ EMS		Adolescents’ EMSs														
	*M* ± *SD*	1. ED 2.02 ± 1.14	2. AB 2.59 ± 1.23	3. MA 2.05 ± 1.07	4. SI 1.88 ± 1.05	5. DS 1.70 ± 0.92	6. FA 1.99 ± 1.12	7. DI 1.75 ± 0.89	8. VH 1.92 ± 1.10	9. EM 1.93 ± 1.07	1. SB 1.93 ± 1.02	11. SS 3.20 ± 1.15	12. EI 2.32 ± 1.23	13. US 3.32 ± 1.16	14. ET 2.64 ± 0.99	15. IS 2.50 ± 1.20
1. ED	2.34 ± 1.10		0.31 **	0.32 **	0.45 **	0.44 **	0.30 **	0.26 **	0.30 **	0.15 *	0.44 **	0.25 **	0.42 **	0.24 **	0.30 **	0.41 **
2. AB	2.18 ± 1.03	0.59 **		0.46 **	0.36 **	0.52 **	0.28 **	0.42 **	0.51 **	0.33 **	0.47 **	0.24 **	0.33 **	0.21 **	0.35 **	0.42 **
3. MA	1.88 ± 0.83	0.45 **	0.40 **	0.18 *	0.51 **	0.51 **	0.23 **	0.32 **	0.51 **	0.39 **	0.37 **		0.37 **	0.19 **	0.35 **	0.35 **
4. SI	1.67 ± 0.73	0.57 **	0.44 **	0.50 **		0.67 **	0.44 **	0.46 **	0.60 **	0.41 **	0.55 **		0.49 **	0.21 **	0.31 **	0.39 **
5. DS	1.53 ± 0.63	0.57 **	0.44 **	0.58 **	0.77 **		0.50 **	0.62 **	0.68 **	0.49 **	0.63 **	0.15 *	0.41 **	0.19 **	0.29 **	0.40 **
6. FA	2.00 ± 0.88	0.46 **	0.43 **	0.53 **	0.66 **	0.73 **	0.17 *	0.60 **	0.48 **	0.30 **	0.56 **	0.19 **	0.32 **		0.16 *	0.37 **
7. DI	1.59 ± 0.70	0.46 **	0.39 **	0.56 **	0.68 **	0.74 **	0.77 **		0.63 **	0.48 **	0.65 **		0.31 **		0.22 **	0.40 **
8. VH	1.58 ± 0.73	0.48 **	0.46 **	0.73 **	0.63 **	0.71 **	0.71 **	0.70 **		0.51 **	0.66 **	0.14 *	0.42 **	0.23 **	0.35 **	0.45 **
9. EM	1.48 ± 0.64	0.50 **	0.39 **	0.58 **	0.58 **	0.65 **	0.64 **	0.72 **	0.69 **	0.20 **	0.45 **		0.25 **		0.23 **	0.29 **
1. SB	1.82 ± 0.71	0.54 **	0.48 **	0.54 **	0.68 **	0.68 **	0.69 **	0.67 **	0.73 **	0.73 **		0.35 **	0.51 **	0.28 **	0.42 **	0.51 **
11. SS	3.31 ± 1.16		0.16 *	0.15 *	0.14 *			0.14 *	0.17 *		0.19 **	0.18 **	0.38 **	0.51 **	0.43 **	0.41 **
12. EI	2.35 ± 0.98	0.45 **	0.36 **	0.47 **	0.56 **	0.51 **	0.49 **	0.54 **	0.54 **	0.52 **	0.57 **	0.37 **		0.32 **	0.49 **	0.54 **
13. US	3.13 ± 1.05		0.17 *	0.29 **	0.18 **	0.26 **	0.21 **	0.23 **	0.29 **	0.16 *	0.27 **	0.52 **	0.36 **	0.30 **	0.57 **	0.34 **
14. ET	2.47 ± 0.86	0.16 *		0.36 **	0.30 **	0.33 **	0.30 **	0.34 **	0.40 **	0.34 **	0.36 **	0.37 **	0.44 **	0.56 **	0.15 *	0.59 **
15. IS	2.40 ± 0.86	0.35 **	0.31 **	0.35 **	0.49 **	0.49 **	0.52 **	0.54 **	0.48 **	0.42 **	0.49 **	0.23 **	0.49 **	0.40 **	0.65 **	0.21 **
16. AIR	4.21 ± 0.71	−0.18 *	−0.14 *	−0.22 **	−0.18 **	−0.20 **	−0.20 **	−0.27 **	−0.17 *	−0.26 **	−0.24 **	0.21 **				−0.15 *
17. AAA	3.93 ± 0.77	−0.18 *		−0.17 *	−0.16 *	−0.14 *	−0.17 *	−0.28 **		−0.23 **	−0.26 **	0.15 *				−0.20 **
18. ASS	4.15 ± 0.74	−0.16 *		−0.22 **	−0.20 **	−0.24 **	−0.22 **	−0.30 **	−0.19 **	−0.29 **	−0.29 **	0.18 **	−0.17 *			−0.23 **
19. AMR	4.11 ± 0.75	−0.21 **	−0.16 *	−0.28 **	−0.20 **	−0.25 **	−0.24 **	−0.30 **	−0.23 **	−0.28 **	−0.29 **	0.22 **	−0.14 *			−0.20 **
20. ASA	4.12 ± 0.69	−0.19 **	−0.15 *	−0.24 **	−0.19 **	−0.21 **	−0.22 **	−0.30 **	−0.19 **	−0.28 **	−0.28 **	0.21 **				−0.20 **

Note: ED = Emotional deprivation, AB = abandonment/instability, MA = mistrust/abuse, SI = social isolation/alienation, DS = defectiveness/shame, FA = failure, DI = dependence/incompetence, VH = vulnerability to harm or illness, EM = enmeshment/undeveloped self, SB = subjugation, SS = self-sacrifice, EI = emotional inhibition, US = unrelenting standards, ET = entitlement/grandiosity, IS = insufficient self-control/self-discipline, AIR = adolescents’ interpersonal relationship, AAA = adolescents’ academic achievement, ASS = adolescents’ social skills, AMR = adolescents’ mental resources, ASA = adolescents’ social adaptation. The correlation on the diagonal is the correlation between parental schema and adolescent counterpart schema. The shaded part is the correlation between adolescents’ EMSs and social adaptation. * *p* < 0.05. ** *p* < 0.01.

**Table 2 behavsci-14-00928-t002:** Independent sample *t*-test for parent–adolescent EMSs.

Schema Domain	EMSs	Parent (*N* = 201)	Adolescent (*N* = 201)	*t*	Father (*N* = 74)	Adolescent (*N* = 74)	*t*	Mother (*N* = 125)	Adolescent (*N* = 125)	*t*
		*M* ± *SD*	*M* ± *SD*		*M* ± *SD*	*M* ± *SD*		*M* ± *SD*	*M* ± *SD*	
DR	1. ED	2.34 ± 1.1	2.02 ± 1.14	2.86 **	2.18 ± 0.96	1.83 ± 1.06	2.06 *	2.44 ± 1.16	2.14 ± 1.18	2.05 *
	2. AB	2.18 ± 1.03	2.59 ± 1.23	−3.59	2.23 ± 1.03	2.46 ± 1.24	−1.26	2.15 ± 1.04	2.64 ± 1.23	−3.38 **
	3. MA	1.88 ± 0.83	2.05 ± 1.07	−1.75	1.96 ± 0.91	1.95 ± 1	0.05	1.83 ± 0.78	2.1 ± 1.11	−2.22
	4. SI	1.67 ± 0.73	1.88 ± 1.05	−2.34	1.63 ± 0.73	1.79 ± 0.96	−1.14	1.71 ± 0.73	1.94 ± 1.11	−1.96
	5. DS	1.53 ± 0.63	1.7 ± 0.92	−2.12	1.45 ± 0.49	1.64 ± 0.77	−1.81	1.58 ± 0.7	1.73 ± 1	−1.39
IAP	6. FA	2 ± 0.87	1.99 ± 1.12	0.12	1.77 ± 0.77	1.92 ± 1.09	−0.99	2.13 ± 0.9	2.03 ± 1.14	0.78
	7. DI	1.59 ± 0.7	1.75 ± 0.89	−1.92	1.51 ± 0.65	1.65 ± 0.78	−1.21	1.64 ± 0.72	1.8 ± 0.95	−1.51
	8. VH	1.58 ± 0.73	1.92 ± 1.1	−3.68	1.49 ± 0.69	1.98 ± 1.18	−3.07 **	1.65 ± 0.75	1.89 ± 1.06	−2.11 *
	9. EM	1.48 ± 0.64	1.93 ± 1.07	−5.08	1.42 ± 0.57	1.91 ± 1.07	−3.48 **	1.52 ± 0.68	1.94 ± 1.08	−3.63 ***
OD	10. SB	1.82 ± 0.71	1.93 ± 1.02	−1.24	1.75 ± 0.7	1.91 ± 0.97	−1.08	1.87 ± 0.71	1.94 ± 1.06	−0.63
	11. SS	3.31 ± 1.16	3.2 ± 1.15	0.90	3.23 ± 1.2	3.18 ± 1.35	0.22	3.35 ± 1.14	3.22 ± 1.02	0.96
OVI	12. EI	2.35 ± 0.98	2.32 ± 1.23	0.31	2.35 ± 1.02	2.29 ± 1.21	0.29	2.36 ± 0.96	2.32 ± 1.26	0.28
	13. US	3.13 ± 1.05	3.32 ± 1.16	−1.75	3.12 ± 1.23	3.23 ± 1.25	−0.53	3.13 ± 0.93	3.37 ± 1.11	−1.86
IL	14. ET	2.46 ± 0.86	2.64 ± 0.99	−1.88	2.46 ± 0.92	2.61 ± 1.01	−0.95	2.48 ± 0.83	2.66 ± 0.98	−1.56
	15. IS	2.4 ± 0.86	2.5 ± 1.2	−0.92	2.37 ± 0.93	2.44 ± 1.19	−0.37	2.42 ± 0.83	2.53 ± 1.21	−0.80

* *p* < 0.05. ** *p* < 0.01. *** *p* < 0.001.

**Table 3 behavsci-14-00928-t003:** Regression analysis of variable relationships in Model 1 (*N* = 201).

Regression Equation	Regression Equation	Overall Fitting Index			Significance of Regression Coefficient	
Outcome Variables	Predictive Variables	*R*	*R* ^2^	*F*	*Β*	*t*
Adolescents’ EMSs	Parents’ EMSs	0.19	0.03	7.02 **	0.25	2.65 **
Adolescents’ social adaptation	Parents’ EMSs	−0.23	0.05	5.53 **	−0.03	−0.37
	Adolescents’ EMSs				−0.21	−3.18 **

Note: ** *p* < 0.01.

**Table 4 behavsci-14-00928-t004:** Regression analysis of variable relationships in Model 2a (*N* = 201).

Regression Equation	Regression Equation	Overall Fitting Index			Significance of Regression Coefficient	
Outcome Variables	Predictive Variables	*R*	*R* ^2^	*F*	*β*	*t*
AMA	PMA	0.18	0.03	6.53 **	0.23	2.56 *
AIR	PMA	0.25	0.06	4.41 **	−0.09	−1.58
	AMA				−0.14	−2.91 **
AAA	PMA	0.18	0.03	3.17 *	−0.05	−0.73
	AMA				−0.12	−2.24 *
ASS	PMA	0.24	0.06	6.13 **	−0.08	−1.32
	AMA				−0.15	−2.96 **
AMR	PMA	0.29	0.08	9.10 ***	−0.08	−1.23
	AMA				−0.18	−3.80 **

Note: AMA = Adolescents’ mistrust/abuse, PMA = parents’ mistrust/abuse, AIR = adolescents’ interpersonal relationship, AAA = adolescents’ academic achievement, ASS = adolescents’ social skills, AMR = adolescents’ mental resources. * *p* < 0.05, ** *p* < 0.01, *** *p* < 0.001.

**Table 5 behavsci-14-00928-t005:** Regression analysis of variable relationships in Model 2b (*N* = 201).

Regression Equation	Regression Equation	Overall Fitting Index			Significance of Regression Coefficient	
Outcome Variables	Predictive Variables	*R*	*R* ^2^	*F*	*Β*	*t*
AIS	PIS	0.21	0.04	9.32	0.30	3.05 **
AIR	PIS	0.18	0.03	3.34	−0.08	−1.36
	AIS				−0.08	−1.86
AAA	PIS	0.23	0.05	5.71	−0.03	−0.43
	AIS				−0.14	−3.18 **
ASS	PIS	0.21	0.06	5.97	−0.08	−1.20
	AIS				−0.15	−2.97 **
AMR	PIS	0.23	0.05	5.46	−0.09	−1.46
	AIS				−0.11	−2.59 *

Note: AIS = Adolescents’ insufficient self-control/self-discipline, PIS = parents’ insufficient self-control/self-discipline. * *p* < 0.05. ** *p* < 0.01. Based on the analysis results of the mediation effect, the specific path coefficients between variables in this study are shown in Figure 2.

## Data Availability

The original contributions presented in the study are included in the article, further inquiries can be directed to the corresponding author.

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
