# Peer review of "How Does Parental Early Maladaptive Schema Affect Adolescents’ Social Adaptation? Based on the Perspective of Intergenerational Transmission"

_behavsci, 2024, doi:10.3390/bs14100928_

Round 1
Reviewer 1 Report
Comments and Suggestions for Authors
The study suggests and affect and intergenerational transmission whereas it is only a survey study. For such strong claims they would need longitudinal research.
Comments on the Quality of English LanguageIt is well written however, I would recommend changing words of affect, effect and any causality since this is a survey study!
Author Response
Comments 1: The study suggests and affect and intergenerational transmission whereas it is only a survey study. For such strong claims they would need longitudinal research. |
Response 1: We changed “Parents’ early maladaptive schemas negatively affects children’s social adaptation indirectly through intergenerational transmission.” to “The results show that adolescents' early maladaptive schema plays an intermediary role between parents' early maladaptive schema and adolescents’ social adaptation.”[ It can be found page 1, paragraph 1, and linen 20-22 in the revised manuscript .] Indeed, longitudinal studies are a better way to determine causality, and we will take such methods into account in our subsequent studies.
|
Comments 2: It is well written however, I would recommend changing words of affect, effect and any causality since this is a survey study! |
Response 2: Agree. In this article, we have adjusted the word "effect" to represent causality, such as: We changed “These results indicate that the self-defeating emotions and cognitive styles formed by parents’ unmet emotional needs do not necessarily directly impact children’s social adaptation, and that the effect of parents’ EMSs on children’s social adaptation is realized through the bridge of children’s EMSs.” to “These results indicate that the self-defeating emotions and cognitive styles formed by parents' unmet emotional needs do not necessarily directly affect children's social adaptation, and the influence of parents' emotional management on children's social adaptation is realized through the bridge of children's emotional management.t.” [ It can be found page 13, paragraph 5, and linen 506-510 in the revised manuscript .] “Further attention should be paid to the negative effect of parents’ lack of self-control/self-discipline schema.”to ” “Further attention should be paid to the negative impact of parents’ lack of self-control/self-discipline schema.” [ It can be found page 14, paragraph 2, and linen 539-540 in the revised manuscript .]
|

Reviewer 2 Report
Comments and Suggestions for Authors
Title of the Paper: "How Does Parental Early Maladaptive Schema Affect Children’s Social Adaptation? An Examination from the Perspective of Intergenerational Transmission."
Thank you to the authors for their significant contribution to advancing scientific research. Here are some recommendations to improve the submitted text and finalize it for publication.
- In the abstract, I suggest including the sociodemographic characteristics of the sample and clarifying the research design. This will assist authors who may wish to use the article for meta-analyses or systematic reviews in the future.
- Provide a general discussion about the developmental tasks of adolescents, particularly focusing on the need for belonging to a group and the role of attachment to parents during this developmental period.
- Further explore the relationship between Early Maladaptive Schemas (EMS) and traumatic experiences, and consider discussing the intergenerational transmission of trauma more extensively (https://doi.org/10.1007/s40653-019-00273-1).
- Clearly define specific hypotheses or alternatively, state (also in the title) that the study is exploratory.
- The term "children" is used, but it might be more appropriate to revise the text by replacing "children" with "adolescents" or "early adolescents." It's important to consider that this is a delicate developmental period, where social references shift from parents to peers, yet parental influence remains strong.
- It is unclear if there is a differentiation between mothers and fathers, if both parents were involved, or if the choice was based on a specific theoretical criterion.
- The cultural discussion outlined is very interesting and could be elaborated further. Additionally, introducing a paragraph to culturally contextualize familial relationships and caregiving practices in the Chinese context within the introduction could be valuable.
- Clarify what is meant by "main caregivers" and discuss gender variables regarding parents in the introduction. This should cover not only early maladaptive schemas but also literature on intergenerational transmission (e.g., trauma, attachment, or schemas if studies are available).
- Better discuss the limitations of the research and provide more guidance for future research.
- Avoid redundancies in the practical section and try to suggest specific strategies for working with adolescents (e.g., Cardoso et al., 2024, in Children—MDPI https://doi.org/10.3390/ijerph21080971).
I look forward to reviewing your article again. I note that the suggested citations are indicative and not mandatory. They are based on my knowledge and the quality of studies but are not binding for future acceptance of the paper. You can replace them with others if preferred.
Good luck!
.
Author Response
|
Comments 1: In the abstract, I suggest including the sociodemographic characteristics of the sample and clarifying the research design. This will assist authors who may wish to use the article for meta-analyses or systematic reviews in the future |
|
Response 1: Thank you for pointing this out. Based on your suggestions, socio-demographic characteristics and research design have been added to the abstract as follows: [ It can be found page 1, paragraph 1, and linen 13-20 in the revised manuscript .] “In this study, cross-sectional design and questionnaire survey were used to collect data to explore the intergenerational influence of early maladaptive schema in families and their relationship with adolescents' social adaptation. The participants were 201 adolescents aged 12 to 16 years and their primary caregivers (father or mother), of whom 125 (62.2%) were boys and 76 (37.8%) were girls. There were 70 fathers (34.8%) and 131 mothers (65.2%). Chinese adolescents and their primary caregivers were surveyed using paired questionnaires, and the Young Schema Questionnaire (short form) and Adolescent Social Adaptation Scale were completed.”
|
|
Comments 2: Provide a general discussion about the developmental tasks of adolescents, particularly focusing on the need for belonging to a group and the role of attachment to parents during this developmental period. |
|
Response 2: Thanks for your suggestion, in response to the study results, we have supplemented the literature evidence to clarify the developmental tasks of adolescents during this period, as well as the effects of needs and attachment to parents during this specific period on the study results. Such as fellows:[It can be found linen page 11, paragraph 2, and 387-399 in the revised manuscript] “4.1. The relationship between parents’ and adolescents’ EMSs The adolescent stage is relatively unstable and easy to change in the individual life with obvious transition. At this stage, the individual's physiology is basically mature, but the psychological and social behavior is in the early stage of development. They are faced with the task of establishing identity and developing independence and autonomy, and the desire to be independent and free from adult control is particularly strong, but they still need to maintain the attachment relationship with their parents, and the peer relationship is gradually developed(Paterson et al.,1995). Independence and attachment have become important contradictions in their development, and the resulting conflict often leads to social adjustment difficulties for adolescents. Previous research results have emphasized the importance of parental attachment style, especially anxiety related to attachment. Importance in predicting the quality of relationships formed between parents and their early adolescent children (Walsh & Zadurian,2022.).
|
Comments 3: Further explore the relationship between Early Maladaptive Schemas (EMS) and traumatic experiences, and consider discussing the intergenerational transmission of trauma more extensively (https://doi.org/10.1007/s40653-019-00273-1). |
|
Response 3: To further explore the relationship between early maladaptive schema (EMS) and traumatic experiences, and to consider a broader discussion of the intergenerational transmission of trauma, we added the following: [ It can be found page 12, paragraph 2, and linen 413-421in the revised manuscript .]
This is consistent with previous research on the intergenerational transmission of trauma, where grandparents' experiences of physical and psychological abuse have been found to increase the risk of sexual and physical harm in the parents' generation, with psychological abuse becoming a particularly critical factor. Not only does it directly impact the parents, but it also increases the risk of sexual victimization for their offspring, thereby exemplifying the familial transmission of psychological trauma. Moreover, this transmission exerts an indirect influence on younger generations, exacerbating their vulnerability to harm, and highlighting the intricate complexity of trauma perpetuation across generations. (Badenes-Ribera et al., 2019)。
|
|
Comments 4: Clearly define specific hypotheses or alternatively, state (also in the title) that the study is exploratory. Response 4: Thank you for your advice. In the study, we have added the content about the research hypothesis and the contribution of the article, as follows: [ It can be found page 3, paragraph 5, and linen 145-152in the revised manuscript .]
“The purpose of this study is to examine the relationship between parents' EMSs, adolescents' EMSs and adolescents' social adaptation, and to fill the gap of empirical research on how EMSs are passed from generation to generation in families and how these EMSs specifically affect adolescents' social adaptation. It is expected to provide a new di-rection for the clinical practice of adolescent family therapy, which can help professionals better understand and intervene in the social adaptation problems of adolescents. The study predicted that: (1) EMSs may be transmitted in the family; (2) Parents' EMSs influences adolescents' social adaptation through mediating role.”
|
|
Comments 5: The term "children" is used, but it might be more appropriate to revise the text by replacing "children" with "adolescents" or "early adolescents." It's important to consider that this is a delicate developmental period, where social references shift from parents to peers, yet parental influence remains strong. |
|
Response 5: Thank you for pointing out this critical issue. We changed the word "children" to "adolescents" in order to be more consistent with the age group of 12-16 years old
|
|
Comments 6: It is unclear if there is a differentiation between mothers and fathers, if both parents were involved, or if the choice was based on a specific theoretical criterion. |
|
Response 6: Thank you for your careful review. Due to factors such as parents' work schedules and differing levels of involvement in child-rearing, there is indeed an imbalance in the ratio of fathers to mothers in the study. However, given the context in China, it is more common for mothers to attend parent-teacher meetings for adolescents, which aligns with the fact that mothers generally take a more active role in child-rearing compared to fathers (Qian et al., 2024). Therefore, we did not intentionally control the gender ratio of parents. Regarding the research sample, since this study focuses on adolescents and their primary caregivers, it includes one student and one parent (i.e., the primary caregiver), meaning the family composition includes only two types: adolescent-father or adolescent-mother. To better clarify the subject matter, we have added relevant explanations. [ It can be found page 4, paragraph 2, and linen 158-159in the revised manuscript .] The details are as follows:
“The adolescent-parent pairs included both adolescent-father and adolescent-mother combinations.”
|
|
Comments 7: The cultural discussion outlined is very interesting and could be elaborated further. Additionally, introducing a paragraph to culturally contextualize familial relationships and caregiving practices in the Chinese context within the introduction could be valuable. |
|
Response 7: Through literature review, we added the influence of family relationship and parenting style on teenagers under Chinese cultural background in the introduction, as follows: [ It can be found page 3, paragraph 3, and linen 123-144 in the revised manuscript .] In traditional Chinese culture, the family is regarded as the core unit of society, while filial piety and family harmony in Confucianism have a profound impact on the interaction patterns among family members (Chuang et al., 2018). This cultural perception is particularly evident in parent-adolescent relationships (Hammond et al., 2015), such as respecting parents and elders, and fulfilling obligations to care for the family. Studies have also found that Chinese adolescents show higher obedience to their parents (Chen et al., 2019; Pan et al., 2021; Zhang et al., 2017). Therefore, Chinese adolescents may be more likely to accept and be influenced by their parents' upbringing, that is, the intergenerational transmission of early adaptive schema occurs. In addition, Chinese parents are less likely to directly show and express love (Deater-Deckard et al., 2011) and less likely to emphasize autonomy (Supple et al., 2009). This has also led to a higher acceptance of excessive control behaviors such as corporal punishment (Wang & Liu, 2018) and strict supervision (Wu et al., 2002) among Chinese adolescents. At the same time, parental psychological control, as a specific manifestation of EMSs in the process of parenting adolescents, can adversely affect adolescents' autonomy and academic achievement, and significantly increase their risk of psychopathological symptoms (Assor et al., 2004; Otterpohl et al., 2019). Therefore, this study aims to explore how the EMSs of parents and adolescents interact in the Chinese context from a cultural perspective, and how these factors affect adolescents' social adaptation. Through this study, we hope to reveal how family care practices unique to China shape adolescent adaptation patterns and provide cultural background support for the formulation of relevant intervention strategies. References: Hammond, R., Cheney, P., & Pearsey, R. (2015). Sociology of the Family. Rocky Ridge Press. http://freesociologybooks.com/Sociology_Of_The_Family/01_Changes_and_Definitions.php Chuang, S., Glozman, J., Green, D., & Rasmi, S. (2018). Parenting and family relationships in Chinese families: A critical ecological approach. Journal of Family Theory & Review, 10, 367-383. Chen, L., Zhang, W., Ji, L., & Deater-Deckard, K. (2019). Developmental trajectories of Chinese adolescents’ relational aggression: Associations with changes in social-psychological adjustment. Child Development, 90, 2153-2170. Pan, B., Li, T., Ji, L., Malamut, S., Zhang, W., & Salmivalli, C. (2021). Why does classroom-level victimization moderate the association between victimization and depressive symptoms? The “healthy context paradox” and two explanations. Child Development, 92 (5), 1836–1854. Zhang, W., Wei, X., Ji, L., Chen, L., & Deater-Deckard, K. (2017). Reconsidering parenting in Chinese culture: Subtypes, stability, and change of maternal parenting style during early adolescence. Journal of Youth and Adolescence, 46(5), 1117–1136. Deater-Deckard, K., Lansford, J. E., Malone, P. S., Alampay, L. P., Sorbring, E., Bacchini, D., Bombi, A. S., Bornstein, M. H., Chang, L., Giunta, L. D., Dodge, K. A., Oburu, P., Pastorelli, C., Skinner, A. T., Tapanya, S., Tirado, L. M. U., Zelli, A., & Al-Hassan, S. M. (2011). The association between parental warmth and control in thirteen cultural groups. Journal of Family Psychology, 25(5), 790–794. Supple, A. J., Ghazarian, S. R., Peterson, G. W., & Bush, K. R. (2009). Assessing the cross-cultural validity of a parental autonomy granting measure: Comparing adolescents in the United States, China, Mexico, and India. Journal of Cross-Cultural Psychology, 40(5), 816–833. Wang, M., & Liu, L. (2018). Reciprocal relations between harsh discipline and adolescents's externalizing behavior in China: A 5‐year longitudinal study. Child Development, 89(1), 174–187. Wu, P., Robinson, C. C., Yang, C., Hart, C. H., Olsen, S. F., Porter, C. L., Jin, S., Wo, J., & Wu, X. (2002). Similarities and differences in mothers’ parenting of preschoolers in China and the United States. International Journal of Behavioral Development, 26(6), 481-491. Assor, A., Roth, G., & Deci, E. L. (2004). The emotional costs of parents’ conditional regard: A self-determination theory analysis. Journal of Personality, 72, 47–88. Otterpohl, N., Lazar, R., & Stiensmeier-Pelster, J. (2019). The dark side of perceived positive regard: When parents’ well-intended motivation strategies increase students’ test anxiety. Contemporary Educational Psychology, 56, 79–90.
|
|
Comments 8: Clarify what is meant by "main caregivers" and discuss gender variables regarding parents in the introduction. This should cover not only early maladaptive schemas but also literature on intergenerational transmission (e.g., trauma, attachment, or schemas if studies are available). |
|
Response 8: Thank you for your advice. In the process of literature review, we did not see any comparative studies on the early maladaptive schema, trauma, attachment and upbringing of the "primary caregiver". This study hopes to make a superficial exploration, but due to the small sample size, the research results need to be further tested in future studies.
|
|
Comments 9: Better discuss the limitations of the research and provide more guidance for future research. |
|
Response 9 Thank you for pointing this out. I/We agree with this comment. Thank you for your pertinent suggestions. At the end of the paper, we explain the limitations of this study in order to provide reference for future research. [ It can be found page 14, paragraph 2, and linen 570-595 in the revised manuscript .] “Despite the results cited, it is important to highlight limitations in this study that should be considered. First of all, cross-sectional data explore only correlation rather than causation. Therefore, it is suggested that future studies use longitudinal study designs to examine these links. Secondly, the sample size of this study is small, which may limit the possibility of identifying the universality of and differences in EMS development due to factors such as adolescents’ different grades and family structure backgrounds. Future studies could expand the sample size and use a wider sample of adolescents of different ages to examine EMSs and adolescents’ social adaptation and detect regularity for earlier prevention and intervention. In addition, as family structure may affect EMSs and social adaptation, future studies could expand the sample size to compare and verify the inter-generational transmission of EMSs in single-parent and two-parent families. In addition, although the results show that parents’ EMSs can influence adolescents’ EMSs and thus their social adaptation, family ecology is very complex, and other factors may play a role in the intergenerational transmission of EMSs. For example, co-parenting and intergenerational parenting are increasingly common in Chinese families, and interaction patterns among family members and satisfaction with family relationships may impact adolescents’ cognitive development and social adaptation; all these factors are worthy of further research. Based on this, the host-object interdependence model considers the non-independence of pair-data, an emerging method of pair-data analysis in marriage, family, and other research fields, and provides an effective means for interpreting pair-relationships. Our study only considered one parent’s influence on adolescents’ EMSs and social adaptation; future research could consider the joint roles played by other important family caregivers and whether adolescents’ upbringing behaviors are not driven only by one parent’s EMSs (as subject-effect), but may be affected by other caregivers’ EMSs (as an object effect). Future research could investigate how family members’ interactions influence adolescents more comprehensively.”
|
|
Comments 10 Avoid redundancies in the practical section and try to suggest specific strategies for working with adolescents (e.g., Cardoso et al., 2024, in Children—MDPI https://doi.org/10.3390/ijerph21080971). |
|
Response 10: Thank you for your helpful advice. We have added a paragraph on the implications for the practices. [ It can be found page 14, paragraph 4, and linen 553-569 in the revised manuscript .] “Our results provide a significant practical reference for cultivating adolescents’ social adaptation from a family system perspective. We propose intervention ideas for improving the social adaptation of adolescents, specifically as follows: 1. Set up classes for parents to help parents understand the formation of early maladaptive schemas, learn scientific parenting styles, provide more support and care for adolescents, and provide a strong guarantee for adolescents to enhance their social adaptability. 2. Use Schema Therapy to intervene and improve EMSs in adolescents. For example, the use of image reshaping technology to help adolescents reshape their emotional responses to early negative experiences, effectively changing the impact of early bad schema on adolescents' emotions and behaviors. Through chair work techniques, adolescents are allowed to engage in dialogue with different parts of themselves, thereby dealing with and changing their early bad schemata, helping adolescents better understand and integrate inner conflicts. These specific strategies not only help adolescents to deal with and improve early maladaptive schema more effectively and enhance their adaptive ability, but also provide practical guidance for clinical work and further optimize intervention effects (Cardoso et al., 2024).”
|

Reviewer 3 Report
Comments and Suggestions for Authors
The present study searches for the intergenerational transmission of Early Maladaptive Schemas (EMSs) between 201 Chinese parents and children. Parents' EMSs can predict similar schemas in their children, which in turn impact the children's social adaptation. On the other hand, the parents' EMSs do not directly affect the children's social adaptation. The effect is mediated through the children's own EMSs. This is an innovative original study and within the scope of the journal. The study provides insights into how maladaptive cognitive schemas may be transmitted across generations and impact adolescent development and adaptation. I suggest several revisions in manuscript.
1. Authors should clearly define the term social adaptation, what they meant, its construct in the first place in introduction.
2. The hypothesis of the study should be provided along with the research questions. They should also clarify the exact contribution of this study, the gap it fills in this section.
3. The procedure for selecting participants is not clear. The method reads, "The researchers sent an email invitation to the selected schools’ moral education directors to ask if they could contact their students and parents for research purposes" (Lines 123-124). However, it is not clear how these schools or cities were selected, any inclusion or exclusion criteria were applied? Or how representative the sample is of that population? These issues should be defined and potential selection bias and the generalizability of the findings should be addressed in limitations and discussed. .
4. The administration of questionnaires is also poorly described. In lines 127—128, it is said that participants could complete the questionnaire over the Internet (Lines 127-128). What happened if they could not do that? Were there any control mechanism to ensure that participants completed the questionnaires independently, without outside influence or distraction (for instance parents impact). How did you to ensure the accuracy and completeness of the responses. Please clarify. Also provide information on length of measures online, the number of screens, and the duration of completing tests.
5. Ethical procedures should be explained in more detail not just mentioning about getting approval from an ethics committee.
6. Why is there imbalance between fathers (70) and mothers (131) in the sample, which could skew results? Were there only one parent from each student in sample, or any other combination? Please define and address these issues.
7. Statistical section seems problematic and needs to be rechecked. Even though authors stated that they provided descriptive statistics, I could not see any information about demographic, clinical or developmental features of students or parents (factors such as socioeconomic status, parental education level, or the child's mental health history that could confound and impact outcome of study) apart from gender and age. These issues should be addressed.
8. The statistical section reads "paired t-test was used to investigate the differences in EMSs between parents and children" (Lines 163-164). However, I could not see a rationale for using a paired t-test, which is typically employed when comparing the means of two related groups (which I believe is not the case here). The use this test, a direct, paired relationship between parents' and children's EMSs should have been present. Please check and revise the analysis.
9. The mediation analysis is briefly described. Details about the steps taken in the mediation analysis should be added to clarify the composition of specific model, to calculate indirect effects, to determine the statistical significance of the mediation effect.
10. The discussion should also focus on the eight schemas that did not show significant correlations between parents and children. Could there be any explanation for this condition? What might be the reasons for that please discuss?
Author Response
Comments 1: Authors should clearly define the term social adaptation, what they meant, its construct in the first place in introduction. |
||||||||||||||||||||||||||||||||||||||||||||||||||||||||||||||||||||||||||||||||||||||||||||||||||||||||||||||||||||||||||||||||||||||||||||||||||||||||||||||||||||||||||||
Response 1: Thank you for your suggestion. The term 'social adaptation' is a crucial concept in this paper, and defining it clearly at the beginning of the introduction will aid readers in better understanding the content of the study. Accordingly, we have incorporated a definition and description of the structure of social adaptation in the introduction, as per your recommendation [ It can be found page 1, paragraph 1, line 30-38 in the revised manuscript .]. The details are as follows: “Social adaptation refers to a state of a harmonious balance between individuals and the social environment by adjusting their physical and mental state through continuous interaction with the living environment (Terziev, 2017). It is the development task of an individual’s whole life, an important target of individual growth and a key indicator of physical and mental health development. From a psychological perspective, social adaptation involves the development of individual coping mechanisms and strategies that enable individuals to manage social challenges and stressors. This can include learning social norms, developing interpersonal skills, and adjusting one's behavior to conform to social expectations (Buss, 2021).”
References: Terziev, V. (2017). Factors affecting the process of social adaptation. European Journal of International Law, 28, 923-935. Buss, D. M. (2021). The Evolution of Human Social Behavior. Oxford University Press.
|
||||||||||||||||||||||||||||||||||||||||||||||||||||||||||||||||||||||||||||||||||||||||||||||||||||||||||||||||||||||||||||||||||||||||||||||||||||||||||||||||||||||||||||
Comments 2: The hypothesis of the study should be provided along with the research questions. They should also clarify the exact contribution of this study, the gap it fills in this section. |
||||||||||||||||||||||||||||||||||||||||||||||||||||||||||||||||||||||||||||||||||||||||||||||||||||||||||||||||||||||||||||||||||||||||||||||||||||||||||||||||||||||||||||
Response 2: Thank you for your advice. In the study, we have added the content about the research hypothesis and the contribution of the article, as follows: [ It can be found page 3, paragraph 5, and linen 145-152in the revised manuscript .]
“The purpose of this study is to examine the relationship between parents' EMSs, adolescents' EMSs and adolescents' social adaptation, and to fill the gap of empirical research on how EMSs are passed from generation to generation in families and how these EMSs specifically affect adolescents' social adaptation. It is expected to provide a new di-rection for the clinical practice of adolescent family therapy, which can help professionals better understand and intervene in the social adaptation problems of adolescents. The study predicted that: (1) EMSs may be transmitted in the family; (2) Parents' EMSs influences adolescents' social adaptation through mediating role.”
|
||||||||||||||||||||||||||||||||||||||||||||||||||||||||||||||||||||||||||||||||||||||||||||||||||||||||||||||||||||||||||||||||||||||||||||||||||||||||||||||||||||||||||||
Comments 3: The procedure for selecting participants is not clear. The method reads, "The researchers sent an email invitation to the selected schools’ moral education directors to ask if they could contact their students and parents for research purposes" (Lines 123-124). However, it is not clear how these schools or cities were selected, any inclusion or exclusion criteria were applied? Or how representative the sample is of that population? These issues should be defined and potential selection bias and the generalizability of the findings should be addressed in limitations and discussed. |
||||||||||||||||||||||||||||||||||||||||||||||||||||||||||||||||||||||||||||||||||||||||||||||||||||||||||||||||||||||||||||||||||||||||||||||||||||||||||||||||||||||||||||
Response 3: Thank you for your suggestion. Perhaps the previous statement was too brief and did not clearly explain the selection, completion and representativeness of the subjects. As one of the more economically developed cities in China, Suzhou, Jiangsu Province, pays more attention to students' mental health, so its school psychology construction is more perfect, and better feedback will be obtained from such a school survey. In addition, teenagers aged 12 to 16 are mainly in middle schools, so we choose middle schools in Suzhou to investigate. Our “Participants and procedures” section restates the selection process of the subjects [ It can be found page 3, paragraph 5, and linen 168-178in the revised manuscript .], as detailed in: “We contacted the ethics director of Suzhou Middle School with the help of the local education bureau and sent them an email application to organize student participation in the investigation. A total of 15 schools accepted the invitation. After the Ethics officer's comments, we held a parents' meeting in the school. During the meeting, parents were introduced to the purpose and procedure of the study, the principle of confidentiality, and the participants' right to terminate the study. During the meeting, the Internet link of the questionnaire was published, and students and parents who agreed to participate in the questionnaire were willing to click the link and fill in the questionnaire. In order to achieve a matching effect between parents and children, parents and children can use their real names or nicknames when answering questions, but they must ensure that both use the same name.”
|
||||||||||||||||||||||||||||||||||||||||||||||||||||||||||||||||||||||||||||||||||||||||||||||||||||||||||||||||||||||||||||||||||||||||||||||||||||||||||||||||||||||||||||
Comments 4: The administration of questionnaires is also poorly described. In lines 127-128, it is said that participants could complete the questionnaire over the Internet (Lines 127-128). What happened if they could not do that? Were there any control mechanism to ensure that participants completed the questionnaires independently, without outside influence or distraction (for instance parents impact). How did you to ensure the accuracy and completeness of the responses. Please clarify. Also provide information on length of measures online, the number of screens, and the duration of completing tests. |
||||||||||||||||||||||||||||||||||||||||||||||||||||||||||||||||||||||||||||||||||||||||||||||||||||||||||||||||||||||||||||||||||||||||||||||||||||||||||||||||||||||||||||
Response 4: Thank you for your attention to detail. First of all, the questionnaire was published during the parents' meeting. In order to avoid the group pressure of parents to participate in the survey, the researcher published two two-dimensional codes at the same time: two-dimensional code 1 was the parents' questionnaire for the research, two-dimensional code 2 was the teenagers' questionnaire for the research, and two-dimensional code 3 was the reading materials of public news. Parents who are willing to participate scan the QR code1 pay attention to the answers, and parents who are unwilling to participate scan the QR code3 Read some public news. After the parents fill in the questionnaire, hand the mobile phone to their children, scan and complete the questionnaire or read the news materials. Secondly, the electronic questionnaire can set mandatory answers, that is, all questions must be completed before submitting the questionnaire. We set all questions as mandatory answers to ensure the integrity of the questionnaire. Finally, the scale contains a total of 97 questions, which takes 15 minutes to answer and is screened three times. In order to explain this problem more clearly, we have made supplements in the "Participants and procedures" section [ It can be found page 4, paragraph 3, and linen 183-195in the revised manuscript .], see details "The questionnaire was published during the parents' meeting. In order to avoid the group pressure of parents to participate in the survey, the researcher published two two-dimensional codes at the same time, two-dimensional code 1 was the parents' questionnaire for the research, two-dimensional code 2 was the teenagers' questionnaire for the research, and two-dimensional code 3 was the reading materials of public news. Parents who are willing to participate scan the QR code1 pay attention to the answers, and parents who are unwilling to participate scan the QR code3 Read some public news. After the parents fill in the questionnaire, hand the mobile phone to their children, scan and complete the questionnaire or read the news materials. All questions contained in the electronic questionnaire are required, that is, all questions must be completed before submitting the questionnaire, so as to ensure the integrity of the questionnaire. The questionnaire lasts 15 minutes."
|
||||||||||||||||||||||||||||||||||||||||||||||||||||||||||||||||||||||||||||||||||||||||||||||||||||||||||||||||||||||||||||||||||||||||||||||||||||||||||||||||||||||||||||
Comments 5: Ethical procedures should be explained in more detail not just mentioning about getting approval from an ethics committee. |
||||||||||||||||||||||||||||||||||||||||||||||||||||||||||||||||||||||||||||||||||||||||||||||||||||||||||||||||||||||||||||||||||||||||||||||||||||||||||||||||||||||||||||
Response 5: Thank you for your reminder. We have added the information regarding the ethical procedures followed in our study. Specifically [ It can be found page 4, paragraph 3, and linen 196-197in the revised manuscript .], it reads: “The study was approved by the Suzhou University Ethics Committee (Approval number: ECSU-2019000189).”
|
||||||||||||||||||||||||||||||||||||||||||||||||||||||||||||||||||||||||||||||||||||||||||||||||||||||||||||||||||||||||||||||||||||||||||||||||||||||||||||||||||||||||||||
Comments 6: Why is there imbalance between fathers (70) and mothers (131) in the sample, which could skew results? Were there only one parent from each student in sample, or any other combination? Please define and address these issues. |
||||||||||||||||||||||||||||||||||||||||||||||||||||||||||||||||||||||||||||||||||||||||||||||||||||||||||||||||||||||||||||||||||||||||||||||||||||||||||||||||||||||||||||
Response 6: Thank you for your careful review. Due to factors such as parents' work schedules and differing levels of involvement in child-rearing, there is indeed an imbalance in the ratio of fathers to mothers in the study. However, given the context in China, it is more common for mothers to attend parent-teacher meetings for adolescents, which aligns with the fact that mothers generally take a more active role in child-rearing compared to fathers (Qian et al., 2024). Therefore, we did not intentionally control the gender ratio of parents. Regarding the research sample, since this study focuses on adolescents and their primary caregivers, it includes one student and one parent (i.e., the primary caregiver), meaning the family composition includes only two types: adolescent-father or adolescent-mother. To better clarify the subject matter, we have added relevant explanations [ It can be found page 4, paragraph 2, and linen 158-159in the revised manuscript .]. The details are as follows: “The adolescent-parent pairs included both adolescent-father and adolescent-mother combinations.”
References: Qian, D., Mi, S., Shi, M., Xia, Y., Xie, T., Xie, Y., & Dong, Y. (2024). A review of fathers’ involvement in children-parenting research. Advances in Psychology, 14(5), 327335. https://doi.org/10.12677/ap.2024.145319
|
||||||||||||||||||||||||||||||||||||||||||||||||||||||||||||||||||||||||||||||||||||||||||||||||||||||||||||||||||||||||||||||||||||||||||||||||||||||||||||||||||||||||||||
Comments 7: Statistical section seems problematic and needs to be rechecked. Even though authors stated that they provided descriptive statistics, I could not see any information about demographic, clinical or developmental features of students or parents (factors such as socioeconomic status, parental education level, or the child's mental health history that could confound and impact outcome of study) apart from gender and age. These issues should be addressed. |
||||||||||||||||||||||||||||||||||||||||||||||||||||||||||||||||||||||||||||||||||||||||||||||||||||||||||||||||||||||||||||||||||||||||||||||||||||||||||||||||||||||||||||
Response 7: Thank you for your advice. In the paper, we add demographic information [ It can be found page 4, paragraph 2, and linen 158-166in the revised manuscript .]: “Forty-eight per cent of participants were from Grade 7, 26% from Grade 8, 13% from Grade 9, 6% from Grade 10, 5% from Grade 11, and 2% from Grade 12. The majority of young people come from rural families (63%). Approximately 21% of participants reported a family monthly income of less than ¥4800 ($687), while 45% had incomes between ¥4800 ($687) and ¥9600 ($1373), 17% between ¥9601 ($1374) and ¥14,400 ($2061), and 17% above ¥14,400 ($2061). The parents' education level was below elementary school (6.0%), middle school (43.9%), and high school or technical school (31.1%). Those with a college degree or above accounted for 19.0%.”
|
||||||||||||||||||||||||||||||||||||||||||||||||||||||||||||||||||||||||||||||||||||||||||||||||||||||||||||||||||||||||||||||||||||||||||||||||||||||||||||||||||||||||||||
Comments 8: The statistical section reads "paired t-test was used to investigate the differences in EMSs between parents and adolescents" (Lines 163-164). However, I could not see a rationale for using a paired t-test, which is typically employed when comparing the means of two related groups (which I believe is not the case here). The use this test, a direct, paired relationship between parents' and adolescents’ EMSs should have been present. Please check and revise the analysis. |
||||||||||||||||||||||||||||||||||||||||||||||||||||||||||||||||||||||||||||||||||||||||||||||||||||||||||||||||||||||||||||||||||||||||||||||||||||||||||||||||||||||||||||
Response 8: Thank you for your professional advice. After re-comparing the meaning of paired t test and independent sample t test, we determined that independent sample t test is more suitable, so we recalculated the data and added it in the paper [ It can be found page8, paragraph 2, and linen300-301 in the revised manuscript .] . See Figure 2 for details Table 2. Independent sample T-test for parent-adolescent EMSs
|
||||||||||||||||||||||||||||||||||||||||||||||||||||||||||||||||||||||||||||||||||||||||||||||||||||||||||||||||||||||||||||||||||||||||||||||||||||||||||||||||||||||||||||
Comments 9: The mediation analysis is briefly described. Details about the steps taken in the mediation analysis should be added to clarify the composition of specific model, to calculate indirect effects, to determine the statistical significance of the mediation effect. |
||||||||||||||||||||||||||||||||||||||||||||||||||||||||||||||||||||||||||||||||||||||||||||||||||||||||||||||||||||||||||||||||||||||||||||||||||||||||||||||||||||||||||||
Response 9: Thank you for your professional advice. We add more about the mediation model in 2.3. Data analytic strategy: [ It can be found page3, paragraph 4, and linen 123-144 in the revised manuscript .] “... Third, refer to the mediation effect test method proposed by Preacher and Hayes' Boot-strap program (Preacher-& Hayes, 2008). Using the Boostrap method in the SPSS Process plug-in (Hayes et al., 2017), the mediation analysis was conducted with the total score of parents' EMSs as the independent variable, the total score of adolescents' EMSs as the mediating variable, and the total score of adolescents' social adjustment as the dependent variable. Fourthly, by further using regression, the dimensions of parents' EMSs are divided into independent variables, the dimensions of adolescents' EMSs are divided into mediating variables, and the total score of adolescents' social adaptation is taken as de-pendent variable, to explore the mediating relationship between adolescents' various types of EMSs and adolescents' social adaptation.”
|
||||||||||||||||||||||||||||||||||||||||||||||||||||||||||||||||||||||||||||||||||||||||||||||||||||||||||||||||||||||||||||||||||||||||||||||||||||||||||||||||||||||||||||
Comments 10: The discussion should also focus on the eight schemas that did not show significant correlations between parents and adolescents. Could there be any explanation for this condition? What might be the reasons for that please discuss? |
||||||||||||||||||||||||||||||||||||||||||||||||||||||||||||||||||||||||||||||||||||||||||||||||||||||||||||||||||||||||||||||||||||||||||||||||||||||||||||||||||||||||||||
Response 10: Thank you for your advice. We stated in the paper on lines 546-555): [ It can be found page 14, paragraph 3, and linen 543-552 in the revised manuscript .]
“That is to say, of the fifteen schemas proposed by Young et al. (2004), thirteen did not affect adolescents’ social adaptation through intergenerational transmission, which may be related to the schema theory that individuals may develop overcompensation strategies when schemas are activated (Young et al., 2004). When parents fail to meet a child’s emotional needs, the child compensates in an opposite way to reduce the pain caused by the schema. For example, a highly-controlled individual may reject all forms of control as an adult and be indulgent in raising their adolescents, giving them all the freedom and love they want. Therefore, it seems that parents’ EMSs are not passed on to adolescents, but the negative impact of over-compensatory coping styles still exist, which can be further explored in future studies.”
|
Reviewer 4 Report
Comments and Suggestions for Authors
The paper focuses on an important topic such as the intergenerational effect of early maladaptive schemas on adolescents’ social adaptation. The study is based on a paired sample of Chinese adolescents and their parents. The design of the study, the methods used in the analysis and the description and discussion of the results are very well presented. The description of limitations of the study is very detail and outlines interesting future directions research.
I have just one suggestion to the authors. It would be interesting and informative for the readers to add a paragraph on the implications for the practices and possibly to reflect on interventions, aiming to support families in which the risk for children and adolescents is heightened.
Author Response
3. Point-by-point response to Comments and Suggestions for Authors |
Response 1: Thank you for your helpful advice. We have added a paragraph on the implications for the practices. [ It can be found page 14, paragraph 4, and linen 553-569 in the revised manuscript .] “Our results provide a significant practical reference for cultivating adolescents’ social adaptation from a family system perspective. We propose intervention ideas for improving the social adaptation of adolescents, specifically as follows: 1. Set up classes for parents to help parents understand the formation of early maladaptive schemas, learn scientific parenting styles, provide more support and care for adolescents, and provide a strong guarantee for adolescents to enhance their social adaptability. 2. Use Schema Therapy to intervene and improve EMSs in adolescents. For example, the use of image reshaping technology to help adolescents reshape their emotional responses to early negative experiences, effectively changing the impact of early bad schema on adolescents' emotions and behaviors. Through chair work techniques, adolescents are allowed to engage in dialogue with different parts of themselves, thereby dealing with and changing their early bad schemata, helping adolescents better understand and integrate inner conflicts. These specific strategies not only help adolescents to deal with and improve early maladaptive schema more effectively and enhance their adaptive ability, but also provide practical guidance for clinical work and further optimize intervention effects (Cardoso et al., 2024).”
|

Round 2
Reviewer 2 Report
Comments and Suggestions for Authors
Thank you!
Reviewer 3 Report
Comments and Suggestions for Authors
Authors responded all of my comments adequately